# Transcriptomic Responses of *Salvia hispanica* to the Infestation of Red Spider Mites (*Tetranychus neocaledonicus*)

**DOI:** 10.3390/ijms241512261

**Published:** 2023-07-31

**Authors:** May Lee, Le Wang, Gen Hua Yue

**Affiliations:** 1Temasek Life Sciences Laboratory, National University of Singapore, 1 Research Link, Singapore 117604, Singaporewangle@tll.org.sg (L.W.); 2Department of Biological Sciences, National University of Singapore, 14 Science Drive, Singapore 117543, Singapore

**Keywords:** plant, agriculture, pest resistance, defense response

## Abstract

*Salvia hispanica* (chia) is a highly nutritious food source and has gained popularity due to its high omega-3 fatty acid content. Red spider mites are a serious problem in the production of *S. hispanica*. However, no study has been conducted to analyze the defensive response to the infestation of red spider mites in *S. hispanica*. To elucidate the molecular mechanisms of the defensive response of *S. hispanica* to red spider mites, we performed a transcriptomic analysis of *S. hispanica* when infested by red spider mites. In the comparative assessment of leaf transcriptomes, a total of 1743 differentially expressed genes (DEGs) were identified between control and mite-infested *S. hispanica*. From these, 1208 (69%) transcripts were upregulated and 535 (31%) were downregulated. The DEGs included transcription factors, defense hormones, and secondary metabolites that were either suppressed or activated in response to spider mite herbivory. Gene Ontology (GO) enrichment analysis revealed that plant secondary metabolites, such as glucosinolates, and signaling pathways, including the jasmonic acid signaling pathway, may play an important role in the defense against red spider mites. This study provides novel insights into the defense response of *S. hispanica* to insect herbivory and could be a resource for the improvement of pest resistance in the chia.

## 1. Introduction

*Salvia hispanica*, commonly known as chia, is an annual herbaceous plant from the family Lamiaceae. It is native to Mexico and northern Guatemala and has been a food source in Mesoamerica since at least 3500 BC [1]. It is a pseudo-cereal, cultivated for its edible seeds, which are rich in α-linolenic acid [2]. Chia is commercially grown and consumed in Mexico, northern Guatemala, Bolivia, Ecuador, Colombia, parts of Australia, and the United States [3,4]. Chia has significant value as a cash crop, as chia seeds have gained popularity in recent years due to their nutritional value and health benefits [5,6]. However, the seed yield is still very low [4], and it is reduced by insects and diseases. To facilitate molecular breeding, DNA markers, including microsatellites, have been identified and used in analyzing genetic diversity and relationships in several cultivars from several countries [4]. The genome [3] and some transcriptomes [7] have been sequenced. These genomic tools set a solid foundation to accelerate the genetic improvement of important traits, such as yield and resistance to pathogens and insects.

The red spider mite (*Tetranychus neocaledonicus*) is a common pest that can affect the production of plants, including *S. hispanica* [8]. These mites feed on the plant’s leaves, causing discoloration, curling, and even death in severe cases. The effects of red spider mites on chia production can vary depending on the severity of the infestation and the stage of plant growth [9]. In the early stages of growth, red spider mites can cause stunted growth and reduced vigor in chia plants. This can result in lower yields at harvest [9]. If the infestation is severe, the mites can cause premature defoliation, which can lead to even lower yields or complete crop failure [8,9,10]. In addition to reducing yields, red spider mites can also affect the quality of chia seeds. Several approaches are available to prevent and control red spider mites in plant production, such as crop rotation, biological controls, and the careful use of pesticides [9]. However, these approaches are tedious and costly. Genetic improvement of pest resistance using conventional selective breeding approaches is long-lasting and complicated [11]. Understanding the molecular mechanisms of the defensive response of plants to pests will help molecular breeding for pest resistance [12]. Unfortunately, to the best of our knowledge, in *S. hispanica*, no study has been performed to understand how this plant responds to the infection of red spider mites. Therefore, the aims of this study were to understand the molecular mechanisms behind the response to the infection of *Tetranychus neocaledonicus* by sequencing, assembling, and annotating the transcriptomes of leaves of normal chia and chia infected by red spider mites.

## 2. Results

### 2.1. Morphological Responses of Leaves to Insect Herbivory

Spider mites were observed feeding on the underside of the leaves of the chia plant. There were obvious and observable morphological changes in the leaves under spider mite attack. In the mite-infested samples, areas of the leaf that had been attacked by the mites were observed to be discolored, forming yellow or white patches (Figure 1A). Following continuous feeding by the mites, the leaves appear bleached and dropped off.

### 2.2. Summary of RNA-Seq

The sequencing of six libraries from the control and mite-infested plants produced a total of 129.1 million high-quality paired-end reads after trimming and filtering. An average of 21.5 million cleaned reads were obtained across all six samples. The controls had higher read coverage compared to the mite-attacked samples (Appendix A). Over 13.3 million reads for each library were uniquely aligned to the reference genome. The plants that were attacked by mites showed a slightly lower uniquely mapping rate compared to the controls (86.3% vs. 90.9%).

### 2.3. Identification of Differential Expression Transcripts

A total of 1743 differentially expressed genes (DEGs) were identified using DESeq2 (Figure 2A and Appendix A) between the control and mite-infested chia. From these, 1208 (69%) transcripts were upregulated and 535 (31%) were downregulated (Figure 2C). PCA of the genome-wide expression patterns of the protein-coding genes between the controls and mite-infested leaves showed substantial differences in expression profiles (Figure 1B). Hierarchical clustering analysis also showed that the DEGs clustered according to control and mite-infested groups (Figure 2B).

### 2.4. Validation of DEGs Using qRT-PCR

For validation of the expression profile of DEGs identified by RNA-seq, 12 genes were randomly selected, and primers were designed on the basis of the corresponding annotated transcript sequences. The comparison of expression patterns in the RNA-seq and qPCR data (Figure 3) showed high consistency, with a correlation coefficient of 0.876 (*p* < 10–15), as examined with Pearson’s correlation test, indicating that the expression profiling of the DEGs determined by RNA-seq is reliable and accurate.

### 2.5. Overall Enriched Gene Ontology (GO) Based on All DEGs

The DEGs between the control and mite-infested transcriptomes were used in enrichment analyses based on the Gene Ontology (GO) terms retrieved against *Arabidopsis* genome annotations (TAIR10) (Figure 4). The most enriched GO terms were defense response to fungus (GO:0050832), phenylpropanoid biosynthesis (ath00940), sesquiterpenoid and triterpenoid biosynthesis (ath00909), response to jasmonic acid (GO:0009753), and innate immune response (GO:0045087), which are related to defense response and immunity. GO terms that were related to cell-wall biogenesis and functions (e.g., GO:0042546 and GO:0009828) were also enriched. Several enrichments related to the production of secondary metabolites such as secondary metabolic process (GO:0019748), isoprenoid metabolic process (GO:0006720), and alpha-linolenic acid metabolism (ath00592) were also found.

From the GO enrichment of upregulated DEGs (Appendix A), the most enriched GO terms were defense response to fungus (GO:0050832), immune system process (GO:0002376), phenylpropanoid biosynthesis (ath00940), and response to oxidative stress (GO:0006979). The most enriched GO terms that were related to the downregulated DEGs (Appendix A) were regulation of molecular function (GO:0065009), tetrapyrrole metabolic process (GO:0033013), photosynthesis—antenna proteins (ath00196), and plant hormone signal transduction (ath04075). These data suggest that plants prioritize defense and immunity over respiration and growth.

### 2.6. Construction of Gene Clusters Showing Associated Expression Patterns

The gene expression clusters based on the expression patterns of all the DEGs were constructed (Figure 5). There were nine clusters in total, each with at least 100 genes. Within each cluster, DEGs showed correlated expression patterns. GO enrichment analysis was performed on clusters 1–3 as significantly enriched GO terms were identified for clusters 1–3, but not for clusters 4–9 (Figure 6 and Appendix A). In clusters 1–3, significantly enriched terms represented defense-related responses. In gene cluster 1, GO terms such as secondary metabolite processes, response to jasmonic acid, response to ethylene, regulation of immune system processes, and phenylpropanoid biosynthesis processes were significantly enriched. In gene cluster 2, only the term jasmonic acid biosynthetic process was significantly enriched, while, in gene cluster 3, the GO terms secondary metabolic process, phenylpropanoid metabolic process, glycosinolate catabolic process, and glucosinolate catabolic process were enriched (Appendix A). We focused on the GO term response to jasmonic acid (GO:0009753) in gene cluster 1 and the individual DEGs in this term (Appendix A). A majority of the DEGs in this GO term were upregulated, including those involved in cell wall modification, signaling, and transcription factors.

The interactions of the enriched GO terms of clusters 1–3 were investigated using network analysis. Five gene network communities were identified (Figure 7A). Enrichment analysis was performed on communities 1–3 (Figure 7B and Appendix A) as significantly enriched terms were identified for communities 1–3, but not for communities 4 and 5. In gene network community 1 (Figure 7B), the genes in the regulatory network were implicated mainly in defense and cellular response to insect herbivory. Gene network communities 2 and 3 (Appendix A) were involved in plant growth and cell communication. We found that most of the group regulators in each network community were transcription factors (TFs) (Figure 8B).

The activation of the Jasmonic acid signaling pathway results in the synthesis of secondary metabolites, such as cyanogenic glucosides [13,14]. Cyanogenic glucosides are phytoanticipins that have been identified in over 2500 plant species. These compounds are recognized for their significant role in plant defense mechanisms against herbivores, due to their bitter taste and the release of toxic hydrogen cyanide upon tissue disruption [15]. Glucosinolates are evolutionarily younger and evolved from cyanogenic glucosides, and the evolutionary connection between cyanogenic glucosides and glucosinolates is supported by similarities in their biosynthesis [15]. Therefore, we focused on the GO term glucosinolate-related metabolic process (GO:0019760) in gene network community 1 (Appendix A). There were 12 DEGs enriched in the GO term. Among them, nine DEGs were upregulated. A majority of the upregulated DEGs were involved in the response to biotic and abiotic stress in glucosinolate-related metabolic processes, such as MYB72, MYB73, and PEN3.

### 2.7. Transcription Factors in Spider Mite Response

A total of 171 transcription factors (TFs) were identified from the Arabidopsis transcription factor database PlantTFDB 5.0 [16] (Appendix A). The differentially expressed TFs were classified into 31 families (Figure 8A and Appendix A). The major families of differentially expressed TFs were WRKY (25), MYB (20), bHLH (19), and ERF (19). There were 27 TF families that had 10 or fewer genes, including NAC (10), G2-like (five), bZIP (four), GRF (three), ARF (two), and E2F/DP (one). Some of these TFs have been reported to be associated with plant responses to biotic stress. A majority (67%) of the TFs, including WRKYs, MYBs, and all the ERFs, were upregulated following spider mite infestation.

## 3. Discussion

In this study, we sequenced transcriptomes of Salvia hispanica leaves in response to spider mite herbivory and determined the DEGs corresponding to the response to insect herbivory. These DEGs and their associated functions were identified to be associated with disease or stress responses to insects.

### 3.1. DEGs in Control and Mite-Infested Samples

Transcriptome sequencing can help to understand the mechanisms behind plant-insect interactions by revealing the genes involved in the response to wounding. qPCR was performed, and the data were compared to the RNA-seq data. High consistency was observed in the expression of the genes between the RNA-seq and qPCR data, showing that the RNA-seq data were reliable.

A total of 1743 differentially expressed genes were identified between the control and mite-attacked leaves. The majority of genes were upregulated in the leaves that were attacked by mites. Genes that were involved in cell-wall modification, such as those that involved the synthesis and deposition of cell-wall components including hemicellulose and pectin, as well as genes that involved secondary cell wall biosynthesis, such as the TBL gene family, were upregulated. The TBL gene family promotes the o-acetylation of cell-wall polysaccharides, which changes the structure and function of the cell wall and influences the resistance of plants against pathogens and insects [17,18]. These structural and biochemical changes in the cell wall could be in response to the spider mite attack [19], as well as to pathogen invasion [19,20] as the cells reinforce their defense. In the spider mite-infested plants, several genes that were involved in defense mechanisms, such as pathogen/bacterial resistance genes, were upregulated. Amongst the downregulated DEGs, there were many genes involved in plant growth. These included ERL2 and HCA2, which are related to cell division, IAA14 and PIN7 which are involved in root development, and genes associated with response to light, such as those encoding photoreceptors, e.g., BAS1, PHOT1, and PHOT2. Plant defensive responses are energetically costly [21], and this downregulation would allow the plant to conserve energy and redirect its resources toward defense-related responses [22].

### 3.2. Pathways Involved in the Response to the Infestation of Tetranychus neocaledonicus

GO enrichment analysis showed that the DEGs that were upregulated were implicated in response to biotic and abiotic stress, such as defense response to fungus (GO:0050832), immune system response (GO:002376), phenylpropanoid biosynthesis (ath00940), response to jasmonic acid (GO: 0009753), and response to ethylene (GO:0009723).

Jasmonates (JA) and its derivatives are involved in a variety of processes, having functions in growth and development, and in response to environmental stresses [23,24,25]. In particular, the JA signaling pathway has been shown to be activated in response to tissue wounding, especially by insect herbivores [26,27,28].

Studies have shown that jasmonate zinc finger inflorescence meristem (ZIM)-domain (JAZ) proteins are the repressors of JA signaling. Under normal conditions, relatively high levels of JAZ proteins inhibit the transcription factors that activate the JA-mediated response [29]. However, during insect attack, the levels of JA increase in the cytoplasm. JA transporters move the cytoplasmic JA into the nucleus, which triggers the interaction of coronatine insensitive 1 (COI1) with JAZ proteins and subsequently leads to the degradation of JAZ proteins to relieve the inhibition to plant growth caused by JA signaling. The degradation of JAZ proteins triggers the release of TFs to activate downstream defense responses [30,31] such as secondary metabolites, which are repellent to herbivorous insects. In our present study, DEGs involving JAZ10 proteins and the JA signaling pathway were upregulated. The upregulation of DEGs involving JAZ10 proteins could suggest that JAZ10 proteins are likely being degraded to activate the transcriptional genes involved in plant defense, allowing the plant to mount a defense response against the insect pests. Conversely, as JAZ proteins are repressors of JA-signaling, the upregulation of the DEGs involving JAZ10 could also serve to terminate the defense response soon after initiation [32]. This might be important in mediating the energy demands of a sustained defense response. The upregulation of DEGs involving JAZ10 could also suggest that the cells are trying to maintain homeostasis, increasing the levels of JAZ proteins after their degradation during plant defense as a response to wound healing. Spider mites have also been known to induce [10] and repress [33] JA response. The upregulation of DEGs that repress the JA response in our study could have been caused by the feeding of spider mites on the chia plant [34]. Certainly, further studies should be conducted to determine if spider mites repress the defense response in chia.

JA and its derivatives increase the production of secondary metabolites after attack by herbivores [14,35]. Plant secondary metabolites play crucial roles in response to environmental stresses and defense against pathogens, herbivores, and insects [36,37]. Cyanogenic glucosides play an important role in plant defense against herbivores due to the release of toxic hydrogen cyanide upon tissue disruption, which gives a bitter taste [15,38]. Here, we observed that the majority of DEGs involved in response to biotic and abiotic stress in the GO term glucosinolate-related metabolic process were upregulated, e.g., PEN3, MYB72, MYB73, and NAC042. Interestingly, these DEGs are involved in responses to pathogens. Plants are susceptible to invasion and infection by microbial pathogens following tissue wounding [39]. In our study, the upregulation of the DEGs in the GO term defense response to fungus (GO:0050832), as well as the glucosinolate-related metabolic process, suggests that the plants were attacked by fungal pathogens following tissue wounding from the insect attack.

### 3.3. TFs Involved in the Response to the Infestation of Tetranychus neocaledonicus

Transcription factors play diverse roles in plant defense responses to biotic stresses by regulating the activation, modulation, and coordination of defense-related gene expression, crosstalk of defense pathways, immune responses, and hormone signaling [40].

We identified 171 transcription factors that were differentially expressed upon the infestation of spider mites. Among them, many TFs were found in the WRKY, MYB, bHLH, and ERF families. This might suggest that WRKYs, MYBs, bHLHs, and ERFs play more important roles in the regulation of defense processes compared to other TF gene families. In our study, a majority of the differentially expressed transcription factors triggered by spider mite herbivory were from the WRKY family. The expression levels of many WRKY genes were significantly upregulated. The WRKY transcription factor family has been associated with responses to biotic and abiotic stress [41]. WRKY TFs play important roles in plant defense signaling pathways such as jasmonic acid and ethylene pathways [42] by directly regulating the expression of JA and ethylene responsive genes. WRKY TFs also confer resistance against pathogens, as well as regulate the expression of pathogenesis-related genes and secondary metabolite biosynthesis genes [43]. In our study, most of the upregulated WRKY TFs were from gene network community 1. The genes in network community 1 were mainly involved in defense. Many of the WRKY TFs that were from gene network community 1 were involved with defense against pathogens. Among them were WRKY18 and WRKY60. WRKY18 and WRKY60 work in a complex with WRKY40 in response to pathogens [44]. WRKY22, which was upregulated in our study, has been found to be activated in response to aphid infestation and play a role in both salicylic acid (SA) and jasmonic acid (JA) signaling [45].

The other TF families that were also similarly affected by spider mite herbivory were MYB (12%), bHLH (11%), and ERF (11%). NAC (6%) and C2H2-type zinc finger (5%) families also responded to spider mite attacks. These TFs are able to be induced by JA signaling and are involved in the response to biotic and abiotic stressors [30].

MYB and NAC TFs are large families of TFs that are involved in regulating various processes such as development and stress responses. MYB TFs play a role in the defense response by regulating the biosynthesis of secondary metabolites such as phenylpropanoids [46] and glucosinolates [47]. The MYB TF family regulates plant growth and development, cell morphology and pattern building, physiological activity metabolism, primary and secondary metabolic reactions, and responses to environmental stresses [48]. NAC TFs are important for regulating responses to drought, salt stress, and pathogen attack [49]. The bHLH TF family plays a role in regulating the genes involved in JA and ethylene signaling and the biosynthesis of secondary metabolites such as glucosinolates and diterpenoid phytoalexins, which confer resistance to herbivores and pathogens [50]. Ethylene response factor (ERF) TFs are involved in regulating the genes involved in biotic and abiotic stress [40], as well as the biosynthesis of secondary metabolites such as alkaloids [50]. They play an important role in defense against necrotrophic pathogens by regulating the expression of genes involved in the biosynthesis of defense metabolites, ethylene signaling pathways, and downstream defense responses [50,51].

## 4. Materials and Methods

### 4.1. Plant Materials

All *Salvia hispanica* plants were grown in a greenhouse in Temasek Life Sciences Laboratory as described in previous papers [3,4]. At 50 days post germination (dpg), six plants were challenged with ~100 adult red spider mites/plant, while other plants were free of red mites. At the age of 65 dpg, the fifth leaf from the top of each of three control plants and three plants affected by *Tetranychus neocaledonicus* was collected, respectively.

### 4.2. RNA Extraction, Library Preparation, and Sequencing

Total RNA was isolated from leaves using the RNeasy Plant Mini Kit (Qiagen, Germany) according to the manufacturer’s protocol. The integrity and purity of the total RNA were assessed with gel electrophoresis. The concentration of each RNA sample was measured using a NanoDrop (Thermo Fisher Scientific, Waltham, MA, USA). Total RNA was treated with RNase-free DNase I (Sigma-Aldrich, Singapore) and used for mRNA library construction with the Illumina TruSeq RNA Library Prep Kit v2 (Illumina, San Diego, CA, USA) in accordance with the manufacturer’s protocol. The prepared libraries were paired-end (2 × 150 bp) sequenced on the Illumina NovaSeq sequencing platform by NovogeneAIT Genomics, Singapore.

### 4.3. Bioinformatic Analyses of RNA-Seq Data

The program process_shortreads in the Stacks package [52] was used to demultiplex the raw reads, filter adapters, and clean up low-quality reads. The cleaned reads were mapped to the reference genome of chia [3], using default parameters, with the program STAR [53]. The expression patterns of annotated genes were analyzed using only uniquely mapped reads. The expression level of each annotated gene was counted with the program HTSeq-count [54] according to the information of the annotated gene features in the genome annotation file [3]. The program DESeq2 [55] was used to normalize the relative expression of transcripts and identify differentially expressed genes (DEGs) across samples.

Transcripts with a count per million (CPM) mapped reads >1 were retained for further analysis. Transcripts with a fold change (FC) value  >2 or <−2 and with a significance value of 0.01 after application of the Benjamini–Hochberg false discovery rate (FDR) were considered differentially expressed genes.

### 4.4. Validation of RNA-Seq Data Using qRT-PCR

Total RNA samples from control and mite-infested samples were digested with DNase I recombinant RNase-free (Roche, Basel, Switzerland) and reverse-transcribed with M-MLV Reverse Transcriptase (Promega, Madison, WI, USA) following the manufacturer’s protocol. A total of 12 primers targeting 12 genes (Appendix A) that were differentially expressed between the control and mite-infested transcriptomes were designed according to the coding sequences obtained from the annotated reference genome [3] using the program Primer3 [56]. Quantitative PCR (qPCR) was carried out on the BioRad CFX96 (Bio-Rad Laboratories, Hercules, CA, USA) using SYBR^®^ Green as a fluorescent dye. qPCR reactions were carried out in triplicate, with each 20 µL reaction volume containing 2.5 µL of 5× diluted cDNA, 0.4 µL (4 µM) of each primer, 10 µL of 2× master mix from KAPA SYBR^®^ FAST qPCR kits (Life Technologies, Carlsbad, CA, USA), and 6.7 µL of sterile water according to the manufacturer’s instructions. Raw data were converted to cycle threshold (Ct) values using software provided by BioRad (Bio-Rad Laboratories, Hercules, CA, USA). The 2^−ΔΔCT^ method was used to quantify the expression level using the glyceraldehyde 3-phosphate dehydrogenase gene (GAPDH) as the housekeeping gene to normalize the relative expression of the genes according to our previous study [57,58]. The fold change of each gene was the ratio of the relative expression of the mite-infested leaf to that of the control leaf. The correlation between transcriptome sequencing (transcripts per million) and qPCR dataset (relative mRNA expression) was assessed by Pearson’s correlation coefficient.

### 4.5. Functional Annotation of DEGs

Gene Ontology (GO) accessions were retrieved for each DEG against the annotation of Arabidopsis using the program Blast2GO [59]. The samples were clustered on the basis of the relative expression of DEGs, using both principal component analysis (PCA) and heatmaps with the program ClustVis [60] to investigate the overall expression patterns between the two groups. The program Metascape [61] was used to carry out gene ontology enrichment analysis and to study protein–protein interaction signaling pathways with Arabidopsis as a reference. Regulatory network analysis was conducted using the program DIANE (Dashboard for the Inference and Analysis of Networks from Expression Data) with default parameters and using Arabidopsis as reference [62].

## 5. Conclusions

We sequenced and annotated the leaf transcriptomes of chia subjected to spider mite herbivory. Plants have a wide arsenal of defensive responses and phytochemicals to deter many herbivores and pathogens, including jasmonic acid signaling and glucosinolates. Understanding the balance between jasmonic acid and salicylic acid signaling pathways could lead to strategies to boost plant defense without compromising growth and development. We showed that TFs play an important role in the Salvia hispanica responses to spider mite herbivory. WRKY, MYB, bHLH, ERF, and NAC TF genes were the most involved in the responses. As the demand for sustainable agriculture grows, an understanding of plant defense responses to pests and pathogens could aid in developing environmentally friendly strategies that reduce the use of chemical pesticides. Identification of crucial genes and pathways involved in nonhost defense can provide valuable insights for developing plant varieties that are more resistant to pests.

## Figures and Tables

**Figure 1 ijms-24-12261-f001:**
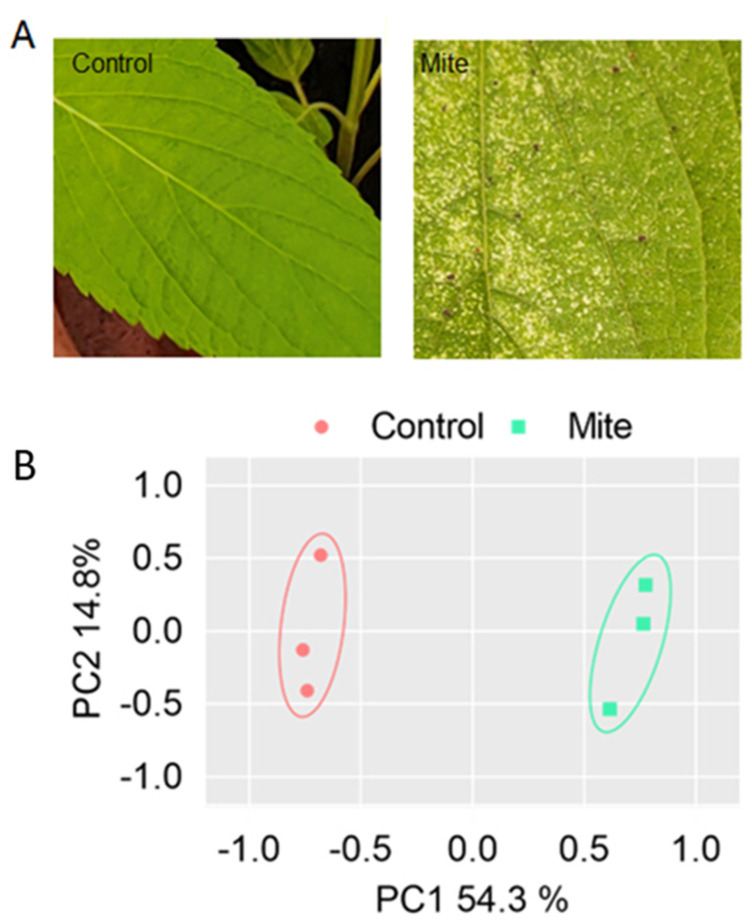
Responses and transcriptome shift of chia leaves to mite feeding: (**A**) phenotypic changes of chia leaves (2× magnification) to mite feeding in contrast to control; (**B**) principal component analysis (PCA) of whole genome-wide gene expression patterns based on transcriptome sequencing between three controlled and three mite-feeding leaves.

**Figure 2 ijms-24-12261-f002:**
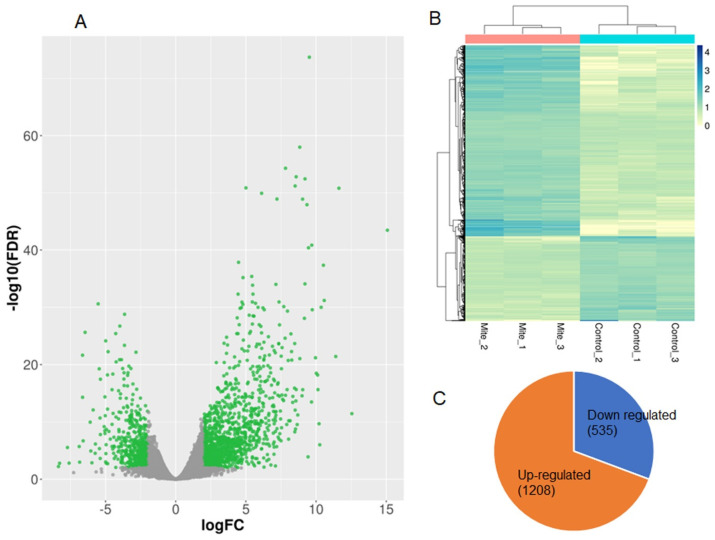
Identification of differentially expressed genes (DEGs) between mite feeding and controls of chia leaf transcriptomes: (**A**) volcano plot of gene expression levels identified by DESeq2, where DEGS are highlighted in green, while the grey color represents genes that are not differentially expressed; (**B**) heatmap of the relative expression levels of the DEGs and the hierarchical relationships of samples between mite-feeding and control samples; (**C**) the number of DEGs and the regulation patterns.

**Figure 3 ijms-24-12261-f003:**
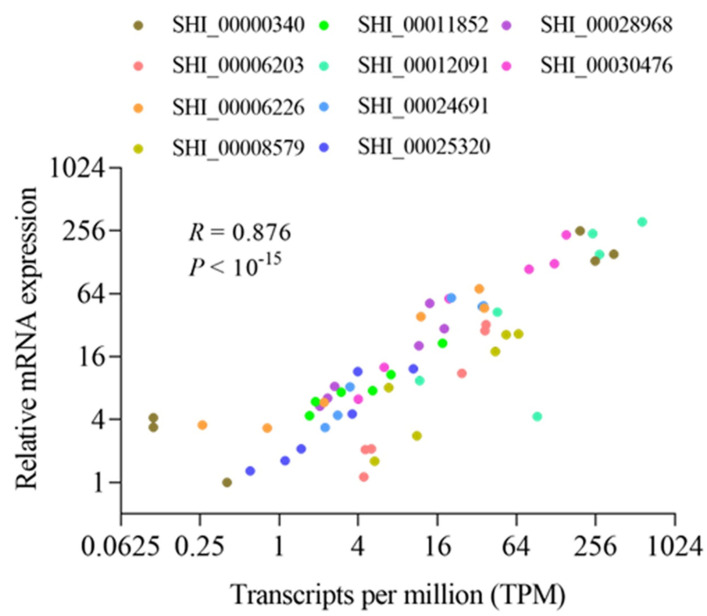
Verification of the randomly selected differentially expressed transcription factors by quantitative real-time PCR (qPCR). The correlation between transcriptome sequencing (transcripts per million) and qPCR dataset (relative mRNA expression) was assessed by Pearson’s correlation coefficient.

**Figure 4 ijms-24-12261-f004:**
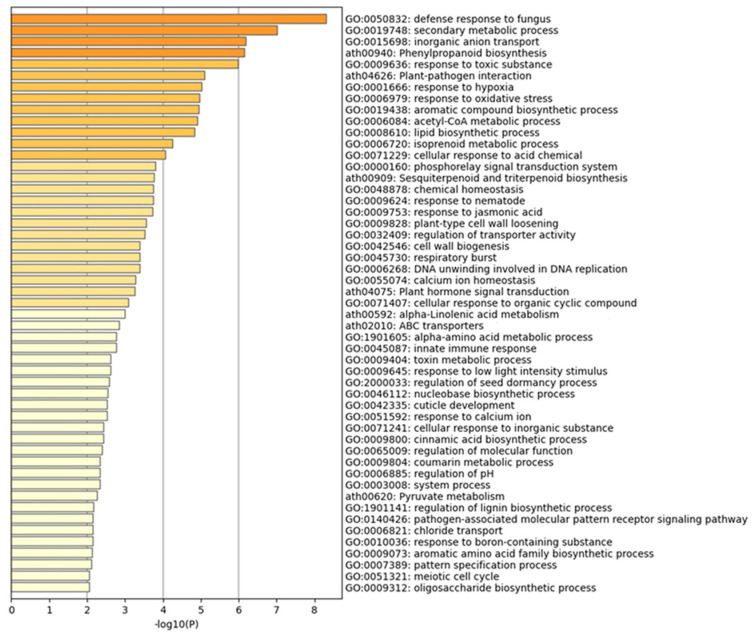
Gene ontology (GO) analysis of differentially expressed genes (DEGs) between mite-feeding and controls of chia leaf transcriptomes at the significance level of 0.01.

**Figure 5 ijms-24-12261-f005:**
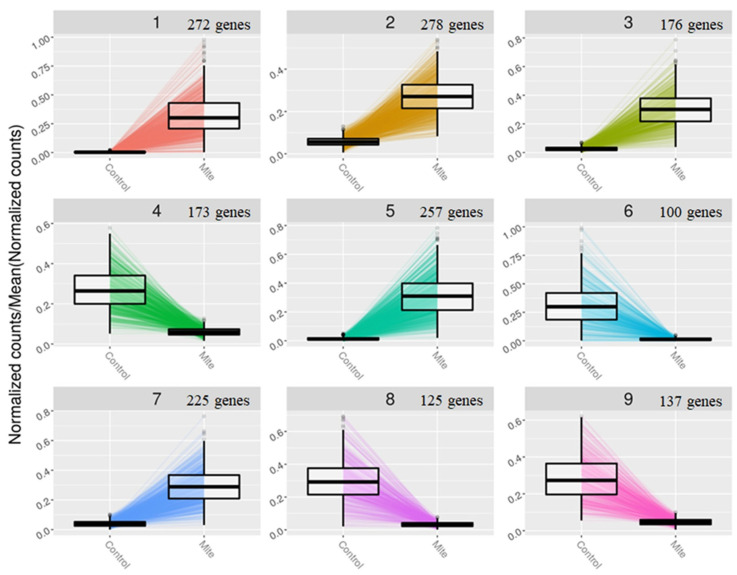
Identification of gene expression clusters (1–9) based on the expression patterns of all the differentially expressed genes (DEGs). The number of DEGs within each cluster is indicated.

**Figure 6 ijms-24-12261-f006:**
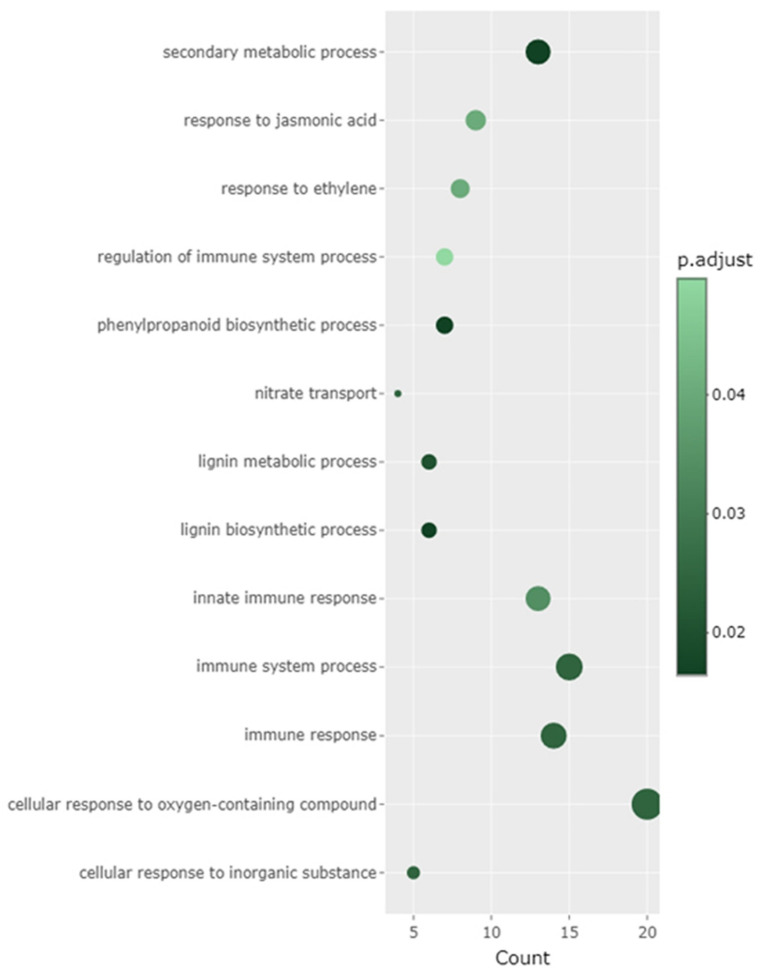
Gene ontology (GO) analysis of differentially expressed genes (DEGs) in gene expression cluster 1. GO terms at the significance level of 0.05 are indicated.

**Figure 7 ijms-24-12261-f007:**
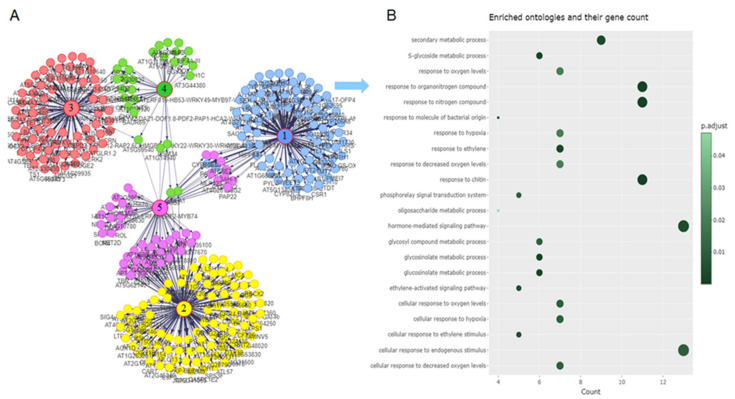
Construction of gene regulation network based on the identified gene expression clusters: (**A**) five gene regulation network communities were identified on the basis of the expression patterns of DEGs. The group regulators of each community are highlighted with circled numbers at the center of each community; (**B**) gene ontology (GO) analysis of differentially expressed genes (DEGs) in gene regulation network community 1. GO terms at the significance level of 0.05 are indicated.

**Figure 8 ijms-24-12261-f008:**
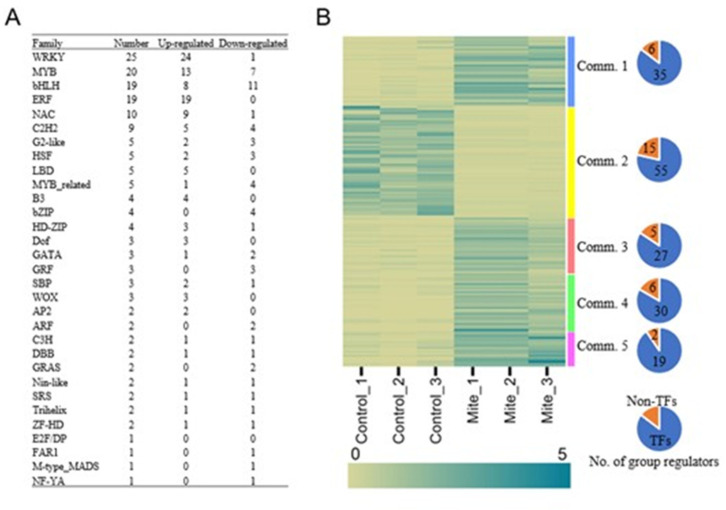
Categorization of differentially expressed transcription factors and the expression patterns between mite-feeding and controls of chia leaf transcriptomes: (**A**) the number of differentially expressed transcription factors in each family; (**B**) heatmap of the relative expression levels of the differentially expressed group regulators in each gene regulation network community, and the number of differentially expressed transcription factors among the total number of group regulators in each gene regulation network community (pie chart).

## Data Availability

Data will be available upon reasonable request.

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
