# Peer review of "Transcriptomic Responses of Salvia hispanica to the Infestation of Red Spider Mites (Tetranychus neocaledonicus)"

_ijms, 2023, doi:10.3390/ijms241512261_

Round 1

Reviewer 1 Report

The manuscript in reference describes the transcriptomic-based differentiation of leaves of Salvia hispanica alone (control) and infested with red spider mites (Tetranychus neocaledonicus), disclosing 1743 differentially expressed genes (DEGs) between control and mite-infested S. hispanica. The manuscript has interesting elements, and the statistical and bioinformatic procedures seem well-performed. However, some relevant points limit its further consideration. The main concern is related to the number of replicates per treatment (i.e., control and mite-infested plants) since three replicates are never not enough to reproduce a clear biological response. This is a severe flaw in this study. In fact, there are some discrepancies since Figure 1B shows three replicates per treatment, and the materials and methods (line 316) indicate six plants to be challenged to the mites (n = ca. 100). Still, only three plants per treatment were selected (criteria?). Although the PCA-derived plot showed statistical differentiation, and probably an induced response, it seems to be over-fitted (explained variance ca. 70%) due to the number of replicates. The number of replicates must be expanded to ensure a biological effect in such a response (>10 replicates) involving a pertinent genetic pool. Therefore, a study with a two-factor design and three replicates is insufficient and cannot define and disclose adequately an effect due to the mite infestation and even elucidate the molecular mechanisms of the defensive response. Indeed, the study design would have been better oriented if it had included plants with known resistance to mites as part of genetic differentiation to better disclose the aim and scope of this study and its hypothesis related to defensive resistance against mites. In this regard, the genotypic variation of test plants and the plant management to control variables are not mentioned or clarified. On the other hand, some results/discussions are over-scoped and highly speculative since there are some misinterpretations. For instance, S. hispanica is a Lamiaceae plant, but the authors argued that glucosinolate can be involved in the defense of this plant. However, Lamiaceae plants can not produce glucosinolates, and the cited study [ref 13] used to support this misinterpreted fact is related to A. thaliana, a Brassicaceae plant, which can produce glucosinolates.

Detailed scrutiny should be performed throughout the manuscript to revise some grammar and stylistic issues.

Reviewer 2 Report

The manuscript by Lee and coworkers describes transcriptome analyses of responses of chia to the infestation of red spider mites. In particular, the authors performed this study in healthy and damaged plants. Analyses of transcriptome identified about 1700 significant DEGs with 69% of them induced by infestation of red spider mites.

This study is well done and well-presented and has generated a significant amount of data which will be very valuable to those working in this area. The authors performed an analysis in transcript expression step and for this reason I think that they showed a preliminary but principal view of gene expression responses to the infestation of red spider mites. On the other hand, the results are largely descriptive and resume results already observed in other species for similar biotic stress responses and without further insight into plant mechanisms and processes, the paper will have limited broader interest.

Just out of curiosity, the authors say “The controls had higher read coverage compared to the mite attacked samples” and at the same time that approximately 2/3 of DEGs are induced by spider infestation. What reasons they give for it?

Round 2

Reviewer 1 Report

The authors adequately addressed and responded to my comments and concerns. In this regard, I'm satisfied with the improvements in the manuscript, so further consideration is recommended.